# Art Definition and Accelerated Experience: Temporal Dimension of AI Artworks

**Wei Liu and Feng Tao ***

College of Philosophy, Nankai University, Tianjin 300071, China
* Correspondence: taofeng@mail.nankai.edu.cn

**Abstract:** Time is a necessary element in understanding AI art. Firstly, time reveals the historical process by which art-theoretical predicates move from the unmarked to the marked, which can thus be utilized as a defense for arguing the legitimacy of AI art as art. Furthermore, AI art should be seen as a "new" art that is temporally ahead of the descriptive forms of art theory. Secondly, time provides a unique interpretation of AI artworks' characteristics and aesthetic experience. The absence of experience, the de-depth of AI artworks, and the "short experience/short memory" aesthetic mode of the masses are closely linked to the scarcity of time in an age of acceleration. Finally, time reflects the loss of "aura" and the new end of art that AI art may bring about. Time provides a defense and explanation of AI art, as well as a perspective for reflecting on the development of art in contemporary times.

**Keywords:** AI art; time; art theory; aesthetic experience; acceleration

## 1. Introduction

The current debate on AI "art" focuses on whether such a thing is possible, in what sense, and, if so, how to position it. Tao Feng believes that AI and art share a common focus on emotion, consciousness, and creativity; in addition, the fact that more and more AI programs are creating art shows that AI art is possible, both in terms of research and practice [1]. Zhao Tingyang asserts that language forms the dialogue between intelligences, and that consciousness occurs in language. If an AI has its own universal language, then the AI may have consciousness [2]. Xu Yingjin argues that in current artificial general intelligence (AGI) research, it is too difficult to achieve the engineering portrayal of intentions and that the NARS system can absorb the reasonable part of the Anscombe intention concept. Moreover, this makes it possible to, therefore, achieve the portrayal of intentions, which is expected to realize the design of an artificial intelligence body with autonomous intentions [3]. Based on the discussion of AI consciousness, there is a tendency to think that AI has not acquired an ontologically independent status and is, thus, unable to create works independently [4]. Furthermore, this notion is then used to assert that AI cannot replace human artists [5] but can only appear as agents. We believe that AI art is one of several possible forms of art in an era of accelerating technology. Moreover, we cannot, therefore, be overly concerned with the technological routes that are being taken in AI art at the expense of thinking about AI art from a temporal perspective. We believe that current art theory does not yet cover the scope of AI art in practice, and that, therefore, there is a need to reflect on art theory in terms of the coordinates of time in order to view it through a more inclusive art-theoretical lens.

In the self-reflection of art theory, time, as an important aspect, has been noted by numerous theorists. Nelson Goodman's "when is art" approach to defining art from the perspective of "art symbols" highlights the co-temporal nature of art and contemporary life [6]. Jerrold Levinson, on the other hand, links the definition of art to the way people move and think, arguing that what art is should be related to the history of art [7]. The core

of his proposal will be an account of what it is to regard-as-a-work-of-art and that which endows this concept an essential historicity. In addition, Arthur Danto refers to the change in artistic predicates in a given time [8]. Although each of these aforementioned efforts have their own respective focus, what they all have in common is that they all show that the question of what art is, as art is somehow linked to time. Blaise Agüera y Arcas argues that the shifts in artistic practice and theory, which were brought about by the technological revolution in photography in the 19th century, can be used in parallel in order to discuss the art of artificial intelligence [9]. Hartmut Rosa, on the other hand, shows that in the age of acceleration, time deprivation has changed the structure of people's experiences [10]. It is easy to see that the current discussion of AI art ignores the role of art theory, and in particular the role of time in art theory.

In this essay, our primary aim is to discuss how the concept of AI art as art is possible through the lens of time, while further providing an analysis of the content characteristics and aesthetic experience structure of AI art. We argue that time provides a rational basis for the renewal of art theory. Firstly, what is considered as art is always guided and limited by art theory, which, in turn, is closely related to the predicates that art theory identifies. However, the predicates of art theory are not constant; they are always subject to—at times significant—changes over time. Therefore, in respect to the criticism of whether AI art is art, it is important to consider how art-theoretical predicates change over time—in other words, if current art-theoretical predicates can accommodate artificiality as a predicate of art, then, surely, there is no problem with regarding AI art as being art. Secondly, time is a key element that not only helps to explain whether AI art is art, but also brings new perspectives to the content and appreciation of AI artworks. We argue that the lack of experience and the de-depthing of AI artworks, as well as the shift in the contemporary aesthetic paradigm of the masses, are closely linked to the lack of time due to the acceleration of society. In short, we believe that time is a necessary element in the discussion of what AI is and what it has to offer. It helps us to understand AI art and also explains part of the problem.

## 2. Time and AI Art in Art Theory

An understanding of AI art cannot be separated from its relationship with traditional art. The central question that is being considered is generally asking whether AI art is art. Whilst we acknowledge the importance of art theory in judging this question, it is important to note that previous debates have been overly conservative in that they have ignored the role of time in art theory. These debates have ignored both the openness of art theory itself to developments over time and the legitimacy of the existence of a new artistic phenomenon—i.e., artificial intelligent art—that has emerged from the practice of art in the light of time. Here, we provide an account of how the temporal element in art theory has a role to play in the legitimacy of considering AI art as art.

Given that AI art is art generated by artificial intelligence, the question generally consists of two issues: (1) how time works when AI-generated works are not considered art; and (2) how time works in the context of AI artists whose purpose is to create art.

In the first case, which can be verified from a broader art historical perspective, we aim to show that throughout the history of art, the time of the day has regularly "shaped" art, and that AI art is only a contemporary case in the development of art history as a whole.

The acceptance of an object as a work of art often takes time to develop. The vast majority of ancient Chinese bronzes, for example, were ceremonial objects that we now regard as works of art [11]. Food, on the other hand, was considered by Kant to be something that provided "possessive pleasure"; therefore, it was not a work of art [12] (p. 42). However, food is now gradually being accepted as an art object [13]. Goodman once suggested that time should not be forgotten when asking the question "what is art?", as he argues that "In crucial cases, the real question is not 'What objects are (permanently) works of art?' but 'When is an object a work of art'" [6] (pp. 66–67).

In much the same way that an object will not always be a work of art, an object will not always remain as not a work of art. The mystery of this notion lies in time. Time sometimes

acts directly on the object itself, not only eroding it but also creating an atmosphere around it. More importantly, time gradually erases the object's original use and intent and provides it with a new one. As utility fades, the object may be seen as a work of art. As in the case of bronze tripods and pottery in museums, their previous practical purpose—i.e., in specific rituals and uses—has been so thoroughly erased by time that the viewer can now only simply admire the forms, patterns, etc., of these artifacts, which thus allows them to become works of art. Here, time becomes a transforming factor, i.e., a transformation of the purpose and intention of the object's production. This is reminiscent of Bullough's "aesthetic distance"—which is the need to maintain a spatial distance in order to appreciate a work of art [14]; however, here, time becomes a distance in another sense as well. The same could be said of AI work. It is conceivable that, in a few years, when we see an AI drawing, we will not associate it with a practical purpose—as an AI drawing mostly to improve its visual recognition—but will simply see it as a painting for the purposes of human enjoyment.

The above example given is a case of AI drawing works that are not intended to generate works of art. In the future, through the passage of time, we will perhaps provide them with the status of works of art. In the following sections, we discuss the second scenario, i.e., in which AI directly imitates human artworks in order to generate works of art, as well as the circumstances under which these works can be regarded as artworks.

Levinson argues that art must be "backward-looking", i.e., that what art is must be placed in the chain of time constituted by art history in order to manifest itself [7] (pp. 232–233). Art transcends time in order to achieve "its inherently continuous evolution" and for people to have an "only satisfying explanation". On the other hand, what art is cannot be arrived at by trying to find the essence of art or by comparing something with some archetype. Levinson states that "The only clue one has is the particular, concrete, and multifarious population that art has acquired at any point" [7] (p. 234). As such, the concept of art has no content beyond what art has been, that is to say, the concept of art is held in common in different places and times and is not an abstract being, but "a concrete conception"—which is to say that it is a concrete activity in which art-aware art makers consciously relate their actions to a pre-existing repository of artworks and produce them in a "regard-as-a-work-of-art" manner.

It is clear that AI art—whether it is defined as imitating human artistic creation or in affirming that it possesses artistic creativity in its own right [15]—is a production from the art already existing in the repository of existing human art. In this way, AI art establishes a connection with human art, but this is not enough for it to become art by itself. It has been argued that there are three periods of time that span between the material object and the artwork. Levinson divides these into the following categories: "$t_p$ (the time of physical creation of the object)"; "$t_i$ (the time of intentioned-object creation)"; and "$t_a$ (the time of art-becoming)" [7] (p. 248). Levinson then discusses these categories in more detail. First is the normal case, i.e., $t_p = t_i = t_a$. This is found in the purpose of a person who produces through their physical form in order to create art within the context of the contemporary artistic period, which is the most common way of creating art. Second is the case of found art, i.e., $t_p > t_i = t_a$. This is typical of the way postmodern art exists, i.e., that the physical form of the artwork predates its artistic form, as in M. Duchamp's work *Fountain*. Third is the case of the naive creator ahead of his time, i.e., $t_p = t_i > t_a$, which means that the physical and conceptual time of the object is ahead of the artistic time that it was produced in. It is within these three temporal modes that a basis is found for discussing the art of artificial intelligence. A reasonable discussion of AI art, however, can only partially agree with Levinson, who, despite his creative ideas about the time-dependent nature of artworks, can run into difficulties when analyzing AI art with his overly focused claim of creator intent.

Therefore, before we delve into the discussion section, we need to make a modification to Levison's third temporal pattern $t_p = t_i > t_a$. For AI art, there are two sets of intentional times: A time when the designer is producing an artistic design (Design time, $D\text{-}t_i$) and a time when the AI art generates art (Generate time, $G\text{-}t_i$). Generally, $D\text{-}t_i > G\text{-}t_i$; as such,

normally, we think of AI art as a program where the designer stores the instructions for the creation of art in a computer, and the program is then run in order to produce the artwork. It is also here that the AI is seen as an AGENT and AI art is seen as an imitation of human art, which, in turn, has led to discussions on issues such as the ownership of the copyright of AI artworks. Therefore, the $D\text{-}t_i > G\text{-}t_i$ time pattern forms the basic premise for thinking about and understanding the above issues, and it is essential to make this distinction.

On the other hand, it is the $D\text{-}t_i \leq G\text{-}t_i$ time pattern that is really challenging. This is not simply to say that the completion of the program design implies the production of an artwork, but rather that the designer is the artificial intelligence program itself. In particular, in the case of $D\text{-}t_i < G\text{-}t_i$ specifically, it implies that the AI first needs to have autonomous consciousness. At the highest level, the programmer has designed only the AI program (which contains some intention of the designer but not necessarily the intention to create art), while the AI is able to create art freely after having an independent consciousness, just as a poet first has the idea to write poetry and then writes it. These two scenarios are discussed together on the basis of the recognition that the AI will have some capacity for self-creation (the latter, in particular, affirming a sense of self-creation and active creation). Although we cannot ignore the fact that doubts about AI art are very common [16], including whether machines have consciousness, it is still notable in the fact that it is being debated to begin with. Having said this, attempts to see whether AI programs can become artists have been ongoing in the early years of AI art development [17], as well as more recently [18].

The distinction of modified temporal patterns reminds us not to fall into the misconception that time is not an important factor for the analysis of AI art, that there is only one form of time in which AI art exists, and that they are never synchronized. It is by ignoring the temporal inconsistencies that researchers criticize the fact that at moment "t", AI art is not the same as human art, and thus, on this basis, suspect the legitimacy of AI art. If we consider time adequately, it is not difficult to see that the temporal structure of AI art is complex and that it can only reasonably be discussed with clear distinctions. It is this discussion that will allow us to return more to AI art as an artistic phenomenon in its own right, while appropriately reducing the disruption of it by scientific and technical difficulties. In the history of art, artists have always created art with the help of various technologies that seemed advanced at the time; as such, technology is not an excuse for art panic and the decline of art. Whether the AI is considered as an agent or as a true creative subject, the simple fact is that the art of AI programs is already widely present in real life. However, the discussion of the legitimacy of AI art needs to be considered, not only in terms of time, but also with the help of a new art theory that can better explain it.

Do the laws of art that have been followed throughout art history mean that the art of the present or the future will inevitably obey them? For AI art, the basic assumptions of criticism are that art has been created by people throughout history and, therefore, all future art will only be created by people; further, it is assumed that the art of the past is so closely tied to the mind that present and future art can hardly be art if it does not address the mind problem. However, is this really the case? A. Danto uses the process of changing predicates in art in order to reveal the inductive errors one can make in the definition of art.

If F and non-F are a set of opposing terms [1] that are also predicates, which are appropriate for describing the relevant artwork K, then there are two possible errors. The first situation is "Throughout an entire period of time, every artwork is non-F. But since nothing thus far is both an artwork and F, it might never occur to anyone that non-F is an artistically relevant predicate. The non-F-ness of artworks goes unmarked" [8] (p. 583). This is the most common error, whereby people leave out certain predicates when using them to describe art. The non-F is not included as a predicate describing art precisely because it is believed that if art is F for a time and all art in the present is F, then non-F will never be a relevant predicate for art. This error is rooted in the neglect of the fact that art predicates may change over time. Even if art is not non-F at one time, it does not mean that art is not non-F at all times. In other words, just because art has been described by mind, as a predicate, before the 21st century does not mean that art will always be described by

mind, nor does it mean that non-mind is necessarily not a predicate for art. It is in this sense that we see the risk of stultifying the art discussion by focusing too much on the "mind" in AI art.

The second case is more specific, and Danto argues that, "All works up to a given time might be G, it never occurring to anyone until that time that something might both be an artwork and non-G; indeed, it might have been thought that G was a defining trait of artworks when in fact something might first have to be an artwork before G is sensibly predicable of it—in which case non-G might also be predicable of artworks, and G itself then could not have been a defining trait of this class" [8] (p. 583). The peculiarity of this situation is that a defining concept of art could be wrong, since it excludes possibilities. If artworks throughout history have always been accompanied by the mind (G), then it is easy to conclude that the mind is a predicate for artworks and that it is the essential characteristic of art, thus giving a defining concept of art that uses the mind to describe art. It may seem that such a definition is reasonable, but where it is wrong is that what is historically an artwork is not necessarily all that is actually or theoretically an artwork, i.e., what is historically called an artwork encompasses far less than what should really be called an artwork. That is to say, the concept of artwork in terms of G does not cover the part of non-G—which is also artwork but is excluded because it is not labeled by us as "artwork" at this current time. Furthermore, the concept of art that always accompanies G is narrow; the true concept of art contains many more, even opposite, artistic predicates, and the definition of art is plural and to some extent even contradictory, which is why Danto says, "G itself then could not have been a defining trait of this class" [8] (p. 583). Do such situations actually exist? Of course they do, and examples can easily be found in art history, such as the contradictory artistic predicates of "reproduction" and "reflection" and the artistic concepts that they define.

Criticism of AI art should be more wary of falling into a second trap. This second case discusses whether the art predicate G at a given time describes the whole of art, or whether art could contain at least some non-G content, or even whether a definition of art predicated on G alone is really reliable. The focus of many researchers' criticisms is how AI art differs from human art. The reason for the difference, it seems to me, is that they only see that G can describe art, and ignore the fact that non-G can work just as well. Therefore, whatever differences they find are aspects of mind, experience, etc., and these differences are not sufficient to show that AI art is not art, but only that it is not human-like art. If human art has always been accompanied by G, then AI art is non-G, and although G and non-G are opposite terms as artistic predicates, they could still exist simultaneously as "art".

It is thought that to make a reasonable assessment of AI art, one cannot ignore the role of the opposite term in the definition of art. If there are *n* related terms of art predicates, then there are also *n* opposite terms of art predicates. The opposite term does not necessarily have nothing to do with art, but rather implies an enrichment of art, "The greater the variety of artistically relevant predicates, the more complex the individual members of the artworld become; and the more one knows of the entire population of the artworld, the richer one's experience with any of its members" [8] (pp. 583–584). It is only with time that the importance of the opposite term becomes apparent. This makes us realize whether the current concept of art is sufficiently inclusive, whether it has a blind exclusivity, and whether it has an innate sense of anthropocentric superiority. Only in this way can we think whether human art and artificial intelligence art should not be easily divided. Does such a divide really exist? Such a distinction, at least in terms of what constitutes an art predicate, shows that art is narrowly defined rather than inclusive. A fuller discussion of the possible forms of art can be achieved by including, rather than excluding, AI art. Does the mere selection of artistic predicates on the basis of human experience clarify all forms of art? The emergence of AI art has given a negative answer to this question. A sound evaluation of AI art includes not only a profound reflection on its characteristics, but also on art theory itself.

Further, we believe that time will reshape the expression of art theory, accommodating new artistic predicates and allowing art to flourish in a new way. For AI art, time will bring at least two changes. At the very least, as AI technology matures over time, AI art will come closer, technically, to what is required in the narrower sense of "art". On the one hand, brain scientists are convinced that consciousness can be reproduced in machines [19] (pp. 55–56), while AI scientists believe that computationalism is still a viable way to achieve machine consciousness [20] (p. 1). Developers of AI art, on the other hand, hope that this type of software will be able "to produce increasingly interesting and culturally valuable pieces of art" and believe that it is only a matter of time before they are recognized as being creative and able to create works of art: "it is our hope that one day people will have to admit that The Painting Fool is creative because they can no longer think of a good reason why it is not" [18] (p. 36).

At the highest limit, time will acknowledge AI art by acting on art theory. Advances in technology may achieve certain difficult breakthroughs (e.g., machine consciousness, creativity, etc.) or they may never be achieved, but none of this matters. What is more important is the way in which time has reshaped the theoretical expression of art. Reinvention manifests itself in three ways. To begin with, new artistic predicates will be encompassed. If the predicates of human art are related to G, the predicates of non-G art should also be taken into account after the emergence of AI art; otherwise, such a conception of art would be banal and old fashioned. When photography was invented, the "technophobic humanists" felt it was an affront to the technique of painting. Walter Benjamin, however, believed that this banal idea of art, which excludes rather than includes, was destined to be abandoned—"the most precise technology can give its products a magical value, such as a painted picture can never again have for us" [21] (p. 510). Benjamin's insights also support the idea of AI art, and the need for artistic concepts to leave the old and accommodate new artistic predicates. Moreover, AI art will not be regarded as art until after art theory has been updated. As Danto said, "to see something as art requires something the eye cannot decry—an atmosphere of artistic theory, a knowledge of the history of art: an artworld" [8] (p. 580). Lastly, art theory should consciously outline the stylistic characteristics of AI art, making it an integral part of the art family. Just as art typically manifested itself in painting and sculpture in the age of crafts, and in photography and film in the age of steam, would art in the age of artificial intelligence not manifest itself in AI art? The relationship between art and the productivity of the times is a matter that art theory urgently needs to explain.

In short, we must also acknowledge that time is not the only element that can make art theory change. Art theory, and what art is, also changes with the practice of art. In this case, art is, as Dickie puts it, "a cultural notion", and it is thus invented in response to the creation of social groups [22] (p. 55). In other words, art was always invented at some point in the past, as was art theory, and it all depends on the process of practice by a social group and when they do it. The work now being done by Aaron (Figure 1), CAN, dalle-2, and Midjourney (Figure 2), among others, suggests that current AI is engaged in the creative activity of art [23]. In particular, recent work produced by the AI tool Midjourney has already won the first prize in the Colorado State Fair's fine arts competition in the United States [24]. From Aaron to Midjourney, the practice of AI art has changed dramatically, and this change demonstrates an inherent span. In other words, time is equally linked to the way art is practiced and can serve as an element to describe the changes in artistic practice. For contemporary art theory, it is important to pay attention to both the abstract changes in the temporal dimension of art predicates and to pay close attention to the development of art practices. More importantly, art theory should be more inclusive in including AI art in the conceptual community of art, as the art world has done with AI art awards.

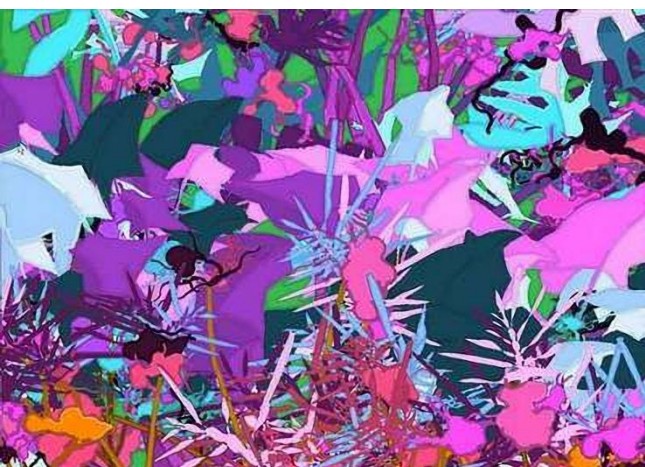

**Figure 1.** Harold Cohen [25], 040601, Pigment on paper, computer-generated, 46 × 57.75 in, 2004.

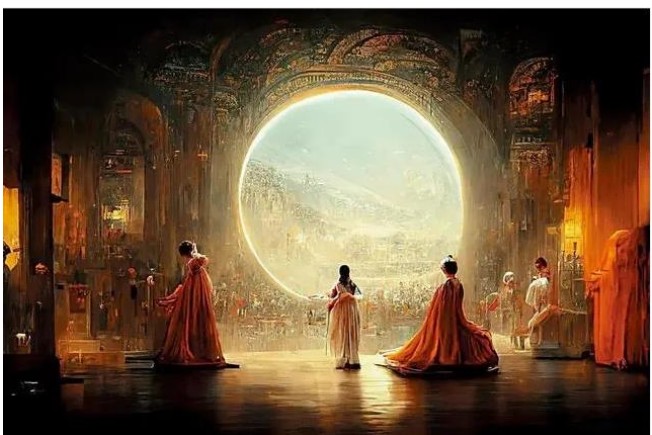

**Figure 2.** *Thétre D'opéra Spatial*. Designed by Midjourney, Won the Digital Art/Digitally-Manipulated Photography Competition First Prize, 2022. Photo from *Ta Kung Pao*.

## 3. Time in the Aesthetic Experience of AI Art

Every aspect of life today seems to be in a state of "acceleration", whether it is the functioning of society, everyday life, or artistic activity. This "acceleration" is almost unstoppable; as such, Paul Virilio points out that the change in speed is the essence of change in the world and that everything is "only dromology" [26] (p. 69). Why? Klaus Dörre explains this as the constant, intrinsic motive force of the capitalist system, the desire of capital to accelerate production in order to maximize profit [27] (p. 146). Art is also being accelerated. This is partly because the technical elements of art can indeed be replaced by faster AI machines, and partly because the reshaped structures of experience of an accelerating age call for new types of art.

The integration of art and technology has a long history, and the emergence of artificial intelligence has accelerated this process. Technology is becoming more and more involved in all aspects of human activity, in the sense that it constitutes the essence of the human being. Bernard Stiegler believes that the deficiencies of human origins can only be remedied by "prosthetics" and that technical prostheses will become part of the body and, indeed, human nature [28] (p. 114). Once we think of technology in this way, we will see that technological proxies are not only "extensions of human organs" but also "simulate the organ they extend" and at the same time "tools are thus 'empirical simulations'" [29] (p. 23). Tools that are extensions of organs not only form part of the body, but also become part of how we experience the world. Therefore, the experience of the world consists of iron picks when people wield them, and the experience of the world consists of computers when

people operate them, which is to say, "body as an evolutionary architecture for operating an awareness in the world. Modify the biological apparatus and you modify its experience of the world" [30] (p. 117). Artificial intelligence, therefore, forms part not only of the body's organs, but also of the world of human experience; further, AI art is the conglomerate of this experience. AI art, thus, heralds a change in the way we experience the world and the way we access aesthetic experiences.

### 3.1. The Experience of Time after Depth Absence

Hartmut Rosa observed that "temporal structures form the central site for the coordination and integration of individual life plans and 'systemic' requirements" [10] (p. 5). Moreover, in real life, people's experience of time is the result of social differentiation and, therefore, changes with society and culture [31] (p. 615). John Tomlinson's study of modernity reveals that acceleration has been the constant leitmotiv of cultural modernity and this, in turn, formed "speed culture" [32]. Further, Caroline Tisdall and Angelo Bozzolla state that "various means of communication, transportation and information have a decisive influence on their psyches" [33] (p. 8). This suggests that social forms and human psychological feelings are in fact in a chain that can be described by time—or, in other words, that feelings and understandings of social culture can also be expressed in feelings about time.

According to Rosa, "the acceleration of the pace of life be defined as the increase of episodes of action and/or experience per unit of time as a result of a scarcity of time resources" [10] (p. 121). When the world of life is accelerated, so is the world of art, and the aesthetic experience of art is invariably accelerated as well. The same is true for AI art and it can be observed both objectively and subjectively.

In regard to objective aspects, AI art has escalated the speed of art in action in three ways. First, the act of creating art itself has accelerated and the process of creating art has become faster. For man, the creation of art depends on "inspiration" (which Plato believed depended on whether the people were possessed by the Muses). However, "inspiration" is not easy to come by, which is why Jia Dao tersely exclaimed that "It took three years to come up with the two verses, and when I recited them, tears streamed down my face" [34] (p. 138). However, for AI, the creative process seems to be completely unconstrained by inspiration and there is little need to wait [35]. Second, rest periods are greatly reduced or, rather, there is almost no need for rest. For any human artist, rest is inevitable. However, machines can work endlessly. Third is multitasking: although the human brain is also multitasking in parallel, artists often need to focus their attention when creating. This means that it is difficult for humans to multitask when creating art. However, for AI, multitasking is already present in a wide variety of scenarios [36], and even when creating art, it does not prevent an AI from working on other problems simultaneously.

Compared to Benjamin's "mechanical reproduction technique", electronic reproduction by AI is not only significantly more efficient but also remarkably different in terms of its content. Benjamin examines the integration of art and technology, arguing that the transcendental distribution of reproduction has undoubtedly increased the speed in the circulation of art, while, at the same time, making the discourse less concentrated on the few [37] (pp. 471–508). While Benjamin notes that the increased speed of art circulation has won opportunities for mass emancipation, he still overlooks the fact that the acceleration of technology brings with it more than a question of equality—technology is not just equal; it is equally unequal. More importantly, according to Rosa, "a growing number of available and potentially interesting goods and pieces of information shortens the span of time that can be devoted to each particular object" [10] (p. 125). Benjamin did not anticipate that AI art was in the same position: the technological acceleration of the art process as a whole is placed in an "acceleration circle" as an event. This not only changes the speed of art circulation, but also results in the appreciators of AI art being wrapped up in the acceleration process and significantly reverses their aesthetic experience. The more efficient the reproduction of art, the more urgent the need for time. Appreciation takes time and patience; however,

patience is at odds with the marketability of efficient circulation. Therefore, appreciation has turned out to be the last link in the chain that prevents circulation. As the intrinsic drive to circulate does not stop, it urges a quickening and superficiality of appreciation. AI can objectively create more artworks, but it also attracts/distracts more of the viewer's attention and time, so almost every work loses its depth and the thickness of interpretation tends to "flatten". The result is that the faster it is produced, the shallower the depth of the AI artwork, the more its value is diluted, and the more people lose patience.

On the subjective side, the perception of time is almost always linked to speed, which is speed = distance/time in physics. In cultural descriptions, there is a similar logic, so that distances become events, and speed is not only movement, but also the speed at which events occur. Thus, the experience of speed is the experience of the event that takes place, it is also the experience of cramming our days with events that place demands on our resources of energy, time, and meaningful belonging [32] (p. 2). According to physics, speed is inversely proportional to time; the faster the speed the shorter the time required. When the speed of technological progress increases, people will need less time to deal with events and will have more leisure and time as a result [38] (p. 1). However, Rosa found people were caught in a time paradox: "time is becoming increasingly scarce despite technical acceleration" [39] (p. 15). Additionally, it seemingly changes the structure of people's experience, "A new form of time experience has emerged that runs counter to the 'classic' 'short experience/long memory' or 'long experience/short memory' model of time experience and time memory, to a 'short experience/short memory' model of time" [10] (p. 94). When AI art accelerates the artistic process, people's expectations and experiences of art are reduced. This may not be due to a lack of depth in AI art itself, but rather to the fact that people are used to a 'short experience/short memory' mode of experience and are unwilling or unable to spend more energy on art content. Kenneth Turan, the Los Angeles Times movie critic, is full of complaints about this: "the summer movies have been tailored more and more to the mindlessness often associated with the tastes of young males"; moreover, "(it) has as much interest in the human condition as a stone" [40] (pp. 5–8). Despite this, these popcorn movies continue to sell well and audiences, as a result, do not care. The reason for this is, in our opinion, that while young audiences may not have lost their taste, time constraints do not allow them to demand more quality content. When people become accustomed to the pay-as-you-go approach to life consumption, they follow the same stereotype in art consumption. In regard to the understanding of artworks, depth means time; therefore, when time is scarce, one loses patience with depth. Art in the age of acceleration is similar to a dessert: too much of it becomes boring but people are happy to try a little bit of everything. The rapid production capacity of AI art satisfies the demand for immediate and increasing varieties in terms of "taste", such that people do not care about depth, but only about the richness of the experience, even if it is a short experience/short memory. In fact, when AI artworks are accused of lacking depth, perhaps only one side of the coin is being seen—in the age of acceleration, depth may become a barrier to appreciation and circulation. AI art "becomes a glorified version of candy crush that seductively maims our bodies and brains into submission and acquiescence" [41] (p. 76). In short, AI art does speed up the art process, possibly to the detriment of the depth of the work; however, it fits the aesthetic needs of a time-poor reality.

### 3.2. Sense of Time after Experience Culling

Benjamin distinguishes two types of experience: the unconscious, emotional accumulation of experience, called the "story", and the conscious noticing of experience, called "information" [42] (p. 88). The difference between the two is that stories are traditionally "chained" and the storyteller has personal experiences attached to the telling of the story, "(like) the way the handprints of the potter cling to the clay vessel" [42] (p. 92). Thus, the story as a link between the individual and the community (i.e., tradition) has an emotional stickiness that builds empathy among the listeners. That is why "a man listening to a story is in the company of the storyteller; even a man reading one shares this companionship" [42]

(p. 100). Information lives only in the "moment" and quickly exhausts itself, such that it is only relevant for the moment and becomes useless after time passes [42] (p. 90). Therefore, in stories, people are always able to mobilize memories about traditions and thus have long-lasting experiences within them. On the contrary, information cuts out experiences and traditions [42] (p. 158), and the brain automatically forgets information that can be found online [43] (p. 15). Thus, in the face of the information bomb, despite all the stimuli one receives, one does not have a long experience of time; rather, they receive only a short experience of time and the accompanying forgetfulness. As such, one is subsequently caught up in the panic of time being silently consumed.

Why is Benjamin's description of experience important to the analysis of the aesthetic experience of AI art? This is due to the fact that the generation of AI art differs significantly from the production of human art, whether it is "(an) application of AI technology in art generation activities", or "an art generation agent that can generate works of art" [44] (p. 1). Additionally, whether it takes the path of symbolism or connectionism, the basis of AI art lies in AI technology. Although AI and computers cannot be equated, the success of AI relies on computers [45] (p. 14). Therefore, while computers can only manipulate symbols, AI art can only be based on the processing of symbols and so the art it creates is not the same as human art in terms of aesthetic experience.

The reason why AI art and human art bring about different aesthetic experiences is due to the different mechanisms of art production. The distinction is similar to that found between a story told by a storyteller and the message conveyed by a newspaper—a different approach, although it may convey the same content. The storyteller is able to share the same story as the listener because the storyteller relies on his experience to tell the story. Although the experience in the story is personal, it is not entirely private; rather, the experience is about tradition [42] (p. 157), such that the experience can call up the memory of its common traditions among different groups. Contrary to this, "(information) to isolate what happens from the realm in which it could affect the experience of the readers . . . Another reason for the isolation of information from experience is that the former does not enter 'tradition'" [42] (pp. 158–159). The inaccessibility to tradition, the inability to transmit private sexual experience and the rapidity with which it exhausts itself are characteristics that make information destined to be inexperienced and barren. Thus, when confronted with human art, we are able to have at least one aesthetic experience, that of a certain memory of tradition. However, we often feel alienated when confronted with AI art, which seems to convey only an aesthetic experience due to the lack of a common memory link; it only presents what the "image" of the work of art conveys, behind which we do not find memories of historical experiences and traditions. This aesthetic experience without history and tradition is also essentially lost in time, but when it becomes an experience in itself, it constitutes our aesthetic experience of the absence of time itself.

However, despite the paucity of experiences of AI artwork, it is still able to "resonate" (resonanz) with the audience. According to Rosa, "resonanz" is not a repetition of the same sound, nor does it last forever, but rather the subject and the world echo each other in their own way, thus requiring a space for the two voices to echo and, thus, establish an empathic relationship [46] (pp. 52–53). In this way, resonance is based on the existence of an echoable space, and the lack of experience with AI art does not mean a lack of resonance with people. Just as Rahel Jaeggi considers alienation as relation of relationlessness [47], inexperience itself becomes a relation, the relation of inexperience—which becomes the basis of empathy between AI art and an equally inexperienced public.

For people living in an accelerated age, everyday life is jam-packed with all sorts of excitement: "the more constantly consciousness has to be alert as a screen against stimuli; the more efficiently it does so, the less do these impressions enter experience" [42] (p. 163). As a consequence, people have only momentary impressions and fragmented experiences, despite appearing to be wrapped up in various perceptual activities. These isolated experiences of time and fragmentation cannot be integrated into a complete structure of experience in which time rushes by, but no content remains in memory or experience. We

spend time in experiences that do not belong to us; we experience all kinds of things but are still very poor in experience. However, the paucity of experience instead becomes a sign, a link that can be echoed between AI art and society at large; when the lack of experience itself becomes a relationship, the viewer is given an echo in AI art. It becomes a temporal experience of disconnection from tradition, of detachment from experience—brought about by the acceleration of technology—which both art and the public have to face.

There is no doubt that this is an alienation of the experience of time. However, Rosa argues that alienation is a prerequisite for resonance, and that resonance is not possible without the initial non-echoing and non-interaction of resonance or, rather, whether the excessive resonance is triggered by a constant echoing and interaction. Therefore, in this vein, a world in which resonance and alienation are dialectically transformed is a world that leads to a good life and sound relationships [46] (p. 317). Likewise, although the lack of experience in AI art is an alienation of experience for human art, an art world that lacks alienation of experience must be inadequate and limp; an aesthetic experience that lacks resonance and an alienation of transformation is equally monotonous and less than beautiful. Thus, we gain grounds for accepting the paucity of time in AI art: the construction of a good life out of the experience of temporal alienation, of which alienation becomes an integral part.

## 4. Conclusions

Danto's art theory explains the possibility of art predicate variation and the impact of predicate variation on artistic judgment. Levinson proposes that artwork has temporal inconsistencies from a temporal perspective, demonstrating that artworks may be ahead of the time frame defined by the art theory of that time. On the one hand, this paper creatively combines Danto's work with Levinson's, arguing that time is an important indicator of changes in art predicates, and that marking changes in art predicates through time highlights the importance of art theory evolving, thereby avoiding artistic conservatism. On the other hand, whilst expanding on Levinson's work, we argue that the emergence of AI art is a typical artistic phenomenon in which artworks are ahead of contemporary art theory, an innovation that has not yet been noted in all previous theories. We argue that while Levinson noted the relationship between art theory and time, its work cannot be directly applied to the interpretation of AI art. For this reason, this paper revises Levinson's theory of the definition of time, in particular, pointing out that an understanding of whether AI art is art requires a distinction between two cases: where the AI does not produce art and where the AI does produce art. The distinction proposed in this paper, i.e., between the time D-ti of AI design and the time G-ti of AI-generated works, effectively avoids critical attention being overly focused on the effects of technological advances. This is achieved while, at the same time, keeping a tight focus on the relationship between art theory and time in thinking about AI art, thereby allowing criticism of AI art to return to the territory of astrology and, more specifically, the relationship between art theory and time.

After discussing whether AI art is art in terms of time, this paper further explains how the production and aesthetic experience of AI art has mutated dramatically as the acceleration of technology has greatly accelerated the pace of life and squeezed the time of the masses. In terms of artistic production, the creation and reproduction of AI are faster and more far-reaching than that of the mechanical age. As time is further squeezed in faster and more distant accelerations, the inherently empirical content of artworks has been further replaced by inexperienced information symbols, resulting in a disconnection from historical tradition and a rupture of personal emotional ties, forming the character of the inexperienced nature of AI artworks. In terms of aesthetic experience itself, the aesthetic structure of the masses has changed dramatically due to a lack of time, whereby there is a tendency to chase short, shallow experiences, a lack of experiential stimulation patterns, and a loss of patience with the pursuit of depth and value in artworks. The faster production capacity, quicker creation methods, and almost inexhaustible creative drive of

AI art cater to the demand for aesthetic experience in a society accelerated by time scarcity, which may not be a wholly good phenomenon, however it is an indisputable fact.

Lutz Koepnick believes that the onslaught of speed brought about by the onset of industrial modernity has opened up a new era of spatial–temporal compression, offering new sensations and stimuli [48] (p. 1). Our sensory systems and aesthetic recognition have changed and it is up to the contemporary artist to jolt the viewer out of familiar perceptual patterns and to make the viewer an active participant in the art of the future.

Although Koepnick's "slowness" is only one suggestion for contemporary aesthetics, he points to the need for contemporary aesthetics to act in the face of acceleration. Not only do artists have to wake everyone up, but contemporary art has an equal responsibility to do so as well. AI art appears to have such potential, and its emergence poses new challenges to aesthetic theory and aesthetic experience. AI art challenges existing art theories, reminding us that changes in art theory cannot ignore the role of time. AI art also challenges traditional ways of experiencing aesthetics and appears to be the beginning of a change in aesthetic interest.

We have always believed that art is, still, conducive to reflection. AI art deepens the question of the end of art. While Danto argues that art ends in narrative, AI art suggests a new approach—which is that AI will gradually replace the replaceable parts of art; that art is moving toward something technologizable. Does all of this, therefore, lead to a new debate on art form and content? On the other hand, the debates about art and technology, which began in Plato's time, and were once again addressed in the age of mechanical reproduction, are now revived and added to by the art of artificial intelligence. Thinking about these issues touches on time as well, albeit with a subtle difference. In short, thinking about AI art in terms of time echoes the demands of aesthetic modernity, and there is still much that is unknown.

**Author Contributions:** Writing—original draft, W.L.; Writing—review & editing, F.T. All authors have read and agreed to the published version of the manuscript.

**Funding:** This research received no external funding.

**Data Availability Statement:** Not applicable.

**Conflicts of Interest:** The authors declare no conflict of interest.

## Notes

[1]　The opposition term has three characteristics: (i) it is different from the contradiction term but can be transformed; the contradiction term requires either a or b, but the opposition term F and -F can be neither. (ii) To be reasonably applicable to something, the object must be of the right sort. For example, yellow and non-yellow as opposites are not suitable to describe the color of water, because water is colorless (but colorless and colored, as contradictory terms, apply to water). (iii) The -F form of the contradictory term is non-specific; further, non-yellow (-F) does not refer specifically to a particular color, but a variety of possibilities.

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
