# Peer review of "Art Definition and Accelerated Experience: Temporal Dimension of AI Artworks"

_philosophies, doi:10.3390/philosophies7060127_

Round 1

Reviewer 1 Report

It’s my pleasure to review this paper. During the review process, I found that there are a few problems in the paper.

The authors put too much emphasis on interpreting previous scholars' opinions, and ignore their own ideas and the logic they want to present to the reader. The most serious problem appears in Section 2. Time and AI Art in Art Theory. The text is not only too long, but also lacks a deep understanding of aesthetics and art theory. The text only focuses on the relationship between time, artwork and art concept, but ignores the evolution of art theory in the practical dimension. Secondly, many opinions and conclusions in the text are too arbitrary and lack the support of references. The author does not pay much attention to the current status of AI art. Too many references related to AI art in the text focus on the period from 2016 to 2019, and lack of sorting and attention to latest typical cases of AI art.

The authors need to make extensive corrections and adjustments to the problems mentioned above. The following are my suggestions for the paper, which I hope will be helpful to the authors.

1.     The current version of abstract is not the core point of the authors' research, but the content quoted from Levinson. It is suggested that the authors elaborate their own point of view, so that readers can better understand the contribution of this research.

2.     In the Introduction section, the authors should write the research background and purpose in a clearer logical way. What are the characteristics and commonalities of time contained in AI art works, and the text lacks the discussion on the relationship between art and AI art .

3.     The authors are suggested to add subtitles in Section 2. Time and AI Art in Art Theory to improve the readability.

4.     The vast majority of ancient Chinese bronzes, for example, were ceremonial objects that we now regard as works of art.”lines 60-61 Where this quote comes from?

5.     Food, on the other hand, was considered by Kant(2015) to be something that provided possessive pleasure, not a work of art, is now gradually being accepted as an art object.”lines 61-63Please consider the correctness of this statement. Did you mean Kant published related materials in 2015? It is a little weird and the relevance of reference 11 is also not clear. Kant is the originator of German classical aesthetics. His aesthetic thoughts are mainly concentrated in Critique of Judgement. The starting point is to combine formal beauty with perfect beauty and put forward "Purposiveness without purpose" as beauty. Time is indeed an important link in Kant's philosophy system, but whether the discussion on time in his aesthetic points is consistent with the main body of this paper needs further discussion.

6.     Lines 314-315, the authors are suggested to revise the reference (Sukegawa Shintaro, et al. 2021) to improve the rigour of the manuscript.

7.     Lines 154-161, the authors mentioned There is a lack of reflection on this issue by many art critics.” What is the basis? 

8.     Lines 162-175, Danto mentioned in his study thatContradictory predicates are not opposites, since one of each of them must apply to every object in the universe, and neither of a pair of opposites need apply to some objects in the universe.” The authors should make better understanding of the “opposites” instead of interpreting them. In other words, understanding the relationship of opposites requires a deeper explanation. For example, the words "black" and "white" both stand for color, but "color" has other colors besides "white" and "black".

9.     In terms of the conclusion section, the authors should explicitly state the novel contribution of this work and its similarities and differences with their previous publications.

Finally, I wish the authors all the very best with this study.

Reviewer 2 Report

Dear Authors, I enjoyed reading your article. The topic of your article is very interesting and relevant. The ideas presented are well argued, and the argumentation is logical.

You have done a very good review of the literature, but you lack research methodology. The goals and objectives of your work are not clearly identified. In the annotation, I would also like to see the purpose of the study

In the work, you also need to clean up some points,

for example, 

remove bolding the beginning of the first letters of the abstract and keywords.

line 360 ​​- a comma at the beginning of the line and others in the text.

Reviewer 3 Report

The paper considers the impact of time on the nature and perception of AI art. While the theme is interesting, especially the consideration of how time influences the reception of artistic works made by AI, at this stage the paper is not ready for publication. The paper is also in need of heavy language revisions.

What the manuscript mainly lacks is structure and a well-developed original argument. The author references past studies on the theme throughout the text but fails to argue for an original position. The impact that time and context have on artworks, detailed in section 2, is a well-addressed topic in the literature and the manuscript in its present form does not bring any new considerations to the debate.

I advise the author to consider which is their main contribution to the discussion and to build a solid argument around that before resubmitting the paper. I would also advise the author to move all the reflections on past literature of the theme in the first part of the paper, to then build on that with an original argument.

Round 2

Reviewer 1 Report

First of all, I affirm the author's revision, but it seems that there is no substantial revision, only a supplementary explanation. Secondly, the author did not follow the revision suggestion: The text only focuses on the relationship between time, artwork and art concept, but ignores the evolution of art theory in the practical dimension.” In addition, although the author has supplemented the relatively new literature, it is not enough to clarify the development status of AI art by adding only one or two of them. It is suggested that the authors should make serious revisions according to the research framework, so that readers can understand the author's research contribution. Finally, I suggest that authors clarify the purpose of their research (lines55-60) and that this research should expound more opinions instead of putting too much emphasis on previous scholars' opinions.

I wish the authors all the very best with this study.

Reviewer 2 Report

Dear authors!

Unfortunately, the article still requires significant elaboration.

1. Abstract.

It should be clear from it what you want to show with your work, why and how relevant it is.

  I recommend that you review other articles for abstract structure.

2. Purpose of your work. Usually it is indicated at the end of the introduction. In your version, at the end of the introduction, something similar to the conclusions sounds.

3. the methodology that you use to achieve the result is still not clear.

I wish you good luck and I hope this is a feasible job.

Reviewer 3 Report

Dear author,

thanks for attentively considering the advice provided. The manuscript in its current form is consistently improved. There are still some typos and language checks that need to be taken care of but overall the paper in its form is now qualified for publication.
